# Fabrication of Magnetic Al-Based Fe_3_O_4_@MIL-53 Metal Organic Framework for Capture of Multi-Pollutants Residue in Milk Followed by HPLC-UV

**DOI:** 10.3390/molecules27072088

**Published:** 2022-03-24

**Authors:** Xue-Li Liu, Yong-Hui Wang, Shu-Yue Ren, Shuang Li, Yu Wang, Dian-Peng Han, Kang Qin, Yuan Peng, Tie Han, Zhi-Xian Gao, Jian-Zhong Cui, Huan-Ying Zhou

**Affiliations:** 1Department of Chemistry, School of Science, Tianjin University, Tianjin 300350, China; liuxueli20190829@163.com; 2Tianjin Key Laboratory of Risk Assessment and Control Technology for Environment and Food Safety, Tianjin Institute of Environmental and Operational Medicine, Tianjin 300050, China; wyhui77@souhu.com (Y.-H.W.); renshuyue2018@163.com (S.-Y.R.); lspla@sina.cn (S.L.); wangyuyu9210@163.com (Y.W.); 15210520025@126.com (D.-P.H.); qinkang2020@foxmail.com (K.Q.); dalidao@139.com (Y.P.); h13601370683@163.com (T.H.); gaozhx@163.com (Z.-X.G.)

**Keywords:** magnetic metal organic framework (MMOFs), magnetic solid-phase extraction (MSPE), multi-pollutants, high-performance liquid chromatography (HPLC)

## Abstract

The efficient capture of multi-pollutant residues in food is vital for food safety monitoring. In this study, in-situ-fabricated magnetic MIL-53(Al) metal organic frameworks (MOFs), with good magnetic responsiveness, were synthesized and applied for the magnetic solid-phase extraction (MSPE) of chloramphenicol, bisphenol A, estradiol, and diethylstilbestrol. Terephthalic acid (H_2_BDC) organic ligands were pre-coupled on the surface of amino-Fe_3_O_4_ composites (H_2_BDC@Fe_3_O_4_). Fe_3_O_4_@MIL-53(Al) MOF was fabricated by in-situ hydrothermal polymerization of H_2_BDC, Al (NO_3_)_3_, and H_2_BDC@Fe_3_O_4_. This approach highly increased the stability of the material. The magnetic Fe_3_O_4_@MIL-53(Al) MOF-based MSPE was combined with high-performance liquid chromatography-photo diode array detection, to establish a novel sensitive method for analyzing multi-pollutant residues in milk. This method showed good linear correlations, in the range of 0.05–5.00 μg/mL, with good reproducibility. The limit of detection was 0.004–0.108 μg/mL. The presented method was verified using a milk sample, spiked with four pollutants, which enabled high-throughput detection and the accuracies of 88.17–107.58% confirmed its applicability, in real sample analysis.

## 1. Introduction

Metal organic frameworks (MOFs) are a kind of porous hybrid material, with a permanent porosity and open crystalline three-dimensional structure, formed by connecting metal ions or metal ion clusters with bridging organic ligands [1,2,3,4]. Compared to other materials, MOFs have the advantages of high porosity, well-defined pore structures, low density, large specific surface area, adjustable pore size, and topological diversity, etc. [5]. However, due to the non-spherical morphology and small particle size of MOFs, their applicability in typical solid-phase extraction (SPE) [6], based on packed sorbents, is hindered. Although SPE can be achieved by dispersing micro/nanomaterial sorbents directly in the sample medium, the recovery of sorbents remains difficult due to the inefficiency of typical centrifugation procedures or filtration [7]. An efficient alternative strategy involves the use of magnetic sorbents to realize easy separation, by applying an external magnetic field from different liquid phases, including the SPE [8].

As a new type of multifunctional composite material, magnetic metal organic frameworks (MMOFs) not only retain the characteristics of high specific surface area and the strong adsorption capacity of MOFs, but also have enhanced magnetism, owing to the introduction of magnetic nanoparticles (MNPs) on the surface or pores of MOFs [9,10,11]. As an adsorbent for magnetic solid-phase extraction (MSPE), MMOFs have shown widespread application prospects in sample pre-treatment [12]. Compared to traditional sample preparation methods, including liquid–liquid extraction [13,14,15], solid-phase extraction [16], and dispersive solid-phase extraction [17,18], MSPE, as a new sample pretreatment technology, has several advantages, such as simple operation, low consumption of organic solvents, and high enrichment efficiency [19,20].

Multi-pollutants are constantly introduced in food, owing to prosperous industrialization, agriculturalization, and anthropic activities. These compounds include different types of chemical substances, such as pesticides [3], pharmaceuticals [21,22], and personal care products [23,24], etc. Because multi-pollutants present potential unknown risks to human beings, it is important to monitor them in food. Multi-pollutant residue methods could provide a useful tool to understand the relationship between pollutant composition and health, as well as provide other advantages, such as a reduction in solvents used, time, costs, and sample. Antibiotics are often used in human and animal medical procedures. As a common antibiotic, chloramphenicol (CAP) was widely used in humans and veterinarians in the past. It had been banned in many countries because of the adverse effects on human health [25]. According to the literature, CAP could still be detected in several food substrates [26], indicating that it was still being used. As a coating on metal packaging materials for food, bisphenol A (BPA) could migrate into food over time and pose a threat to human health, even under suitable storage conditions. As typical environmental endocrine disruptors, estradiol (E2) and diethylstilbestrol (DES) could disrupt the normal function of the human endocrine system after ingestion. In conclusion, CAP, BPA, E2 and DES, as different types of typical pollutants in food, may coexist in food and have adverse effects on human health. Therefore, we selected these substances for simultaneous detection. At the same time, it embodied the advantages of high-throughput for HPLC.

Herein, we proposed Fe_3_O_4_@MIL-53(Al) as an MSPE adsorbent for multi-pollutant residues (Figure 1). Firstly, the Fe_3_O_4_ magnetic core was synthesized and then it was coated with silica to protect the Fe_3_O_4_ core, so that the structure of the material would not be damaged, even in strong acids, strong bases and organic solvents. This also facilitated the reuse of material. Subsequently, amino modification could not only increase the functional group of the material, which was conducive to the adsorption of the target analyte; however, amino modification facilitated amidation between amino and carboxyl groups in organic ligands during the synthesis of materials. Finally, terephthalic acid (H_2_BDC) organic ligands were pre-coupled on the surface of amino-Fe_3_O_4_ composites (H_2_BDC@Fe_3_O_4_). Fe_3_O_4_@MIL-53(Al) MOF was fabricated by the in-situ hydrothermal polymerization of H_2_BDC, Al (NO_3_)_3_, and H_2_BDC@Fe_3_O_4_. A series of characterizations were performed on the synthesized MMOFs. Four pollutants, viz. CAP, BPA, E2, and DES, were chosen as model analytes, and HPLC was used to study the adsorption performance of MOFs and to optimize the parameters that affected the adsorption efficiency. Under optimal conditions, the method was successfully applied to detect multi-pollutants in milk samples.

## 2. Results and Analysis

### 2.1. Characterization of Fe_3_O_4_@MIL-53(Al)

The morphologies of Fe_3_O_4_@MIL-53(Al) and the intermediate were characterized using SEM. As illustrated in Figure 2a, the amino-modified Fe_3_O_4_ magnetic materials had a uniform particle size, with a diameter of approximately 50 nm. As shown in Figure 2b, when H_2_BDC organic ligands were coupled on the surface of amino-modified Fe_3_O_4_, significant morphological changes on the surface could be observed. Tiny rod-shaped crystals were observed when excess H_2_BDC was used. After the hydrothermal reaction, as shown in Figure 2c, MIL-53(Al) presented a cubic structure, with a size of approximately 1 μm, which was intricately connected with magnetic nanoparticles. SEM-mappings can analyze the element type and concentration in the micro-region of materials. As shown in Figure 2d–i, O, Fe, and N were distributed uniformly in MIL-53(Al). A significant increase in C and Al contents was observed because of the high concentration ratio of C and Al in MIL-53(Al).

The XRD patterns of the H_2_BDC-modified magnetic material and Fe_3_O_4_@MIL-53(Al) are displayed in Figure 2j. The black line represents the characteristic diffraction peak of amino-modified Fe_3_O_4_ magnetic materials. When 2θ = 30.12°,35.6°,43.2°,53.4°,57.1° and 62.7°, the characteristic diffraction peak of the amino-modified Fe_3_O_4_ magnetic materials of Fe-O can be clearly observed [27]. The red line represents the characteristic diffraction peak of the H_2_BDC-modified magnetic material. When the coupling reaction occurs successfully, a characteristic peak, different from that of the magnetic ball, can be observed at 2θ ≈ 20°. This indicated the success of the coupling. The blue line represents the characteristic diffraction peak of Fe_3_O_4_@MIL-53(Al). The intensity of the diffraction peak of the amino-modified Fe_3_O_4_ was weakened, and the peaks observed at approximately 9.4°, 10.7°, 12.4°, 16.7°, 17.9°, and 21.9° were attributed to MIL-53(Al) [28].

FT-IR spectra of the synthetic materials are exhibited in Figure 2k. The peak at 574 cm^−1^ is the characteristic absorption peak of Fe-O, indicating the successful preparation of Fe_3_O_4_ nanoparticles. The peak at 1096 cm^−1^ corresponds to the Si-O bond, and that which 3438 cm^−1^ corresponds to is attributed to the vibration of Si-OH on the surface of nano-SiO_2_, indicating that SiO_2_ was formed on the surface of Fe_3_O_4_ nanoparticles [29]. The absorption band at 1631 cm^−1^ corresponds to N-H, indicating the successful amination. In the red line, the absorption peak at 1730 cm^−1^ can correspond to amide bond Ⅰ, while the weak peak, at approximately 1300 cm^−1^, is attributed to C-N. In addition, the band at 3420 cm^−1^ for the intermediate was connected to N-H in the amide. Different characteristic peaks were observed for Fe_3_O_4_@MIL-53(Al). The characteristic peak at 595 cm^−1^ is attributed to Al-O from MIL-53 [30], while the bands at 1411 cm^−1^ and 1591 cm^−1^ are attributed to the symmetrical and asymmetrical stretching vibrations of COO-, respectively [31]. The broad peak at 3434 cm^−1^ corresponds to O-H in MIL-53 [32]. All these results demonstrated that the preparation of the composite materials was successful.

The magnetic properties of the synthetic materials were studied by vibrating sample magnetometry (VSM), and the hysteresis curves are shown in Figure 2l. The saturation magnetization value was 20.68, 19.56, and 13.20 eum g^−1^ for amino-modified Fe_3_O_4_, H_2_BDC-modified Fe_3_O_4_, and Fe_3_O_4_@MIL-53(Al), respectively. The decrease in magnetic saturation was due to the addition of non-magnetic H_2_BDC and Al-MOF during the synthesis. However, in the presence of an external magnet, the material mixed in the aqueous solution can still be separated quickly, within 30 s (Figure 2l). Therefore, Fe_3_O_4_@MIL-53(Al) was easily separated and recycled during the subsequent experiments.

### 2.2. Optimization of the Experimental Parameters

The MSPE processes for the enrichment of CAP, BPA, E2, and DES by Fe_3_O_4_@MIL-53(Al), mainly included adsorption and elution procedures. Some important parameters, such as adsorption time, adsorbent dosage, pH of the sample, and desorption solvent, were optimized to achieve optimal enrichment efficiency.

#### 2.2.1. Effect of Adsorption Conditions

In the MSPE program, pH is a crucial factor, because it can affect the present state of the target analytes [32,33]. In this study, pH was varied from 4 to 11, and as shown in Figure 3a, most pollutants achieved the highest recovery rate at pH = 7. Hence, sample solutions at pH = 7 were used in the following experiments.

During the adsorption process, one of the most vital factors is time. The adsorption time affects the adsorption balance between adsorbent and analyte [34]. In this study, the adsorption time was varied from 2 to 30 min. The study was carried out with increments of 5 min after 2 min, and as shown in Figure 3b, the extraction of target analytes was very fast: more than 70% adsorption occurred within the first 2 min, and equilibrium was almost reached after only 5 min. Therefore, adsorption time was fixed at 5 min in future experiments.

In the MSPE process, when the extraction efficiency is the highest, the adsorption reaches equilibrium. In this study, the amount of adsorbent was varied over the range of 0.5–25.0 mg [35], and as shown in Figure 3c, the recovery rate increased rapidly and reached its maximum at 2.0 mg. However, to ensure a sufficient amount of material and good dispersibility in the solution, we finally chose 25 mg of the material.

#### 2.2.2. Optimization of Desorption Conditions

During the elution process, elution solvents, such as ethanol, acetonitrile and methanol, were studied, and the elution time was evaluated to determine the optimal conditions. As shown in Figure 3d, when the elution solvent was acetonitrile, most of the analytes reached the maximum recovery rate and the analytes dispersed quickly in acetonitrile. After 5 min of desorption (Figure 4a), there was no significant increase in the desorption capacity. Therefore, the optimum elution conditions were as follows: elute with acetonitrile (1 mL, 0.5 mL each time) for 5 min (Appendix A: Appendix A).

### 2.3. Possible Extraction Mechanisms

In general, the pH of the solution was related to the interaction between the adsorbent and the target. First, during the optimization of pH conditions, the adsorption efficiency changed with the pH value of the solution. Because the adsorption efficiency depended on the pH, electrostatic interactions could occur between the adsorbent and the target. Second, considering the structure of the MOF and targets, the metal center of MIL-53 was the hydrophilic center, and the benzene ring of the organic linker was the hydrophobic center [36]. At the same time, there were benzene rings in the structure of the targets (Appendix A); therefore π–π stacking interaction could also occur to promote adsorption [37]. Meanwhile, the resultant magnetic composite contained amino functional groups, which results in hydrogen bonding between the adsorbent and the targets. In conclusion, considering the existence of electrostatic interaction, hydrogen bonding interaction and π–π stacking interaction, the material had super adsorption efficiency for the four targets.

### 2.4. Reusability

Cyclic extraction and desorption tests of the target analyte by the Fe_3_O_4_@MIL-53(Al) adsorbent were conducted to examine the reusability of the adsorbent. As shown in Figure 4b, after 10 adsorption and desorption cycles, the extraction recovery rate remained almost the same, indicating that the prepared material had good reusability.

### 2.5. Method Validation

The quantitation of CAP, BPA, E2 and DES was performed with standard calibration. As shown in Appendix A, excellent linearities were obtained in the range of 0.05–5.00 μg/mL with R^2^ ≥ 0.9989. (Appendix A) The LOD (S/N = 3) and LOQ (S/N = 10) of the proposed method were in the ranges of 0.004–0.106 μg/mL and 0.008–0.209 μg/mL, respectively. The intra-day and inter-day precisions (RSDs) were 0.12–0.79%, and 1.02–1.24%, respectively.

The practicability of the method was evaluated by determining the target substance in the milk sample under optimal conditions. The spiking recoveries of the analytes were obtained by adding the analytes at three different concentrations of 0.10, 0.15, and 0.20 μg/mL. The experimental data are presented in Table 1. Three types of EDCs and CAP were not detected in the empty milk samples. The recovery rate was 88.17–113.46% and the RSD was 0.002–1.951%. These results indicated good reproducibility of the proposed method. The magnetic composite Fe_3_O_4_@MIL-53(Al) showed a good recovery rate, not only in skimmed milk but also in whole milk, which further demonstrated the utility of the material in the complex food matrix. The HPLC patterns of the actual sample are shown in Appendix A.

A comparison between the proposed method and reported methods was conducted to further evaluate the effectiveness of the proposed method. As shown in Table 2, the recovery rates and RSDs of the method developed in this study are either analogous to or better than those of previously reported methods. Moreover, compared to most reported methods, our method requires less adsorbents and requires less time to reach adsorption equilibrium.

## 3. Materials and Methods

### 3.1. Reagents and Chemicals

Ferric chloride hexahydrate (FeCl_3_⋅6H_2_O), ferrous sulfate heptahydrate (FeSO_4_⋅7H_2_O), aluminum nitrate nonahydrate (Al (NO_3_)_3_⋅9H_2_O), *N*-(3-dimethylaminopropyl)-*N*′-ethylcarbodiimide hydrochloride (EDC), *N*-hydroxysuccinimide (NHS), tetraethyl orthosilicate (TEOS), ethanol, *N*,*N*-dimethylformamide (DMF), and ammonia solution were supplied by Aladdin Biochemical Technology Co., Ltd. (Shanghai, China). Further, 3-Aminopropyltriethoxysilane (APTES) was supplied by Yuan ye Bio-Technology Co., Ltd. (Shanghai, China). Hydrochloric acid was provided by Sinopharm Chemical Reagent Co., Ltd., (Shanghai, China), and terephthalic acid (H_2_BDC) was purchased from Xushuo Bio-Technology Co., Ltd. (Shanghai, China). Thermo Fisher Scientific (Waltham, MA, USA) offered HPLC-grade acetonitrile and methanol. All reagents were analytical grade unless stated otherwise. Ultrapure water (Millipore, Burlington, MA, USA) with a specific resistance of 18.2 MΩ was used in all experiments.

All standards (purity ≥98%), including CAP, BPA, E2, and DES, were obtained from Aladdin Industrial Corporation (Shanghai, China). Each standard solution at a concentration of 1 mg/mL was prepared in HPLC-grade methanol. These standards were stored in brown glass bottles at 4 °C. The working standard solutions were prepared daily by diluting the stock solution with ultrapure water.

### 3.2. Instruments and Chromatographic Conditions

The FT-IR spectra were collected using a Nicolet 6700 spectrometer (Thermo Fisher, Waltham, MA, USA) in the wavelength range of 400–4000 cm^−1^. X-ray diffraction (XRD) analysis was performed on a Rigaku diffractometer (Smart Lab 9KW, Tokyo, Japan) using Cu Kα radiation (45 kV, 100 mA). The scans were performed at diffraction angles ranging from 5° to 90°. The morphology and elemental composition of the nanocomposite were obtained using SEM (Merlin compact, Zeiss, Oberkochen, Germany). SQUID-VSM (Quantum Design) was used to study the magnetic properties of the material.

Thermo Ultimate 3000 Liquid Chromatography instrument (Waters Alliance, Milford, MA, USA) with an autosampler and a photodiode array (PDA) detector (Waters, Milford, MA, USA) was used for the analysis of multi-pollutants. The detection wavelengths of PDA were set at 220 nm, 230 nm, and 278 nm. Chromatographic separation was carried out on a SunFireC18 analytic column (250 mm × 4.6 mm, 5 μm; Waters, Milford, MA, USA) using an isocratic elution. The method was carried out in an isocratic mode with a mobile phase composed of Water:Acetonitrile (45%: 55%) at 1.0 mL/min flow rate. The injection volume was 10.0 μL and the column temperature was set at 30 °C.

### 3.3. Sample Preparation

As a food of animal origin, milk is prone to contamination. In addition, milk has high protein content and complex matrix, so we chose it as the sample representative to verify the practicality of this method. The most common pure milk on the market usually included whole milk and skimmed milk. The two types of milk were different in fat and partial nutritional composition, and whole milk was more complex in composition. Milk (whole milk and skimmed milk) was supplied by a supermarket in Tianjin, China. The samples were prepared according to the previously reported methods [41]. A 1.5-mL milk sample was placed in a 7-mL tube and the same volume of acetonitrile was added. After rotational oscillation for 20 min, the samples were centrifuged at 12,000 rpm for 10 min. The supernatant was blown with N_2_ to make it almost dry, and the volume was adjusted to 1.5 mL with ultrapure water for further MSPE analysis.

### 3.4. Preparation of Fe_3_O_4_@MIL-53(Al)

Amino-modified Fe_3_O_4_ was synthesized as reported previously [42]. First, 5.21 g of FeCl_3_·6H_2_O and 4.22 g of FeSO_4_⋅7H_2_O were added into 250 mL ultrapure water and stirred to dissolve completely and then pumped and filtered 850 μL HCl was added before ultrasonically deoxygenating for 30 min. Ammonia solution was added while stirring to adjust the pH to 10. The solution was stirred at 80 °C, 500 rpm for 40 min. The precipitate was separated from the reaction medium by magnetic separation repeated washing with ultrapure water and absolute ethanol. The material obtained in the first step was dispersed into 500 mL absolute ethanol and 250 mL ultrapure water. Next, 38 mL NH_3_·H_2_O was added drop by drop and then 50 mL TEOS was added into it. After stirring for 3.0 h at 60 °C, the magnetic nanomaterials were washed as in the previous step. Then it was dispersed in 200 mL ethanol. Finally, APTES was added for amino functionalization 154 mL APTES was added into 200 mL ethanol with the magnetic nanomaterials. After stirring for 10 h at 75 °C, the aminated magnetic materials were obtained and washed with ultrapure water and absolute ethanol and dried in vacuum at 50 °C. To fabricate of Fe_3_O_4_@MIL-53(Al), 172 mg of H_2_BDC, 100 mg of amino-modified Fe_3_O_4_, 400 mg of EDC and 100 mg of NHS were mixed with 5 mL of ultrapure water. The mixture was oscillated for 2 h keeping away from the light at room temperature, followed by separation using an external magnet, and addition of Al (NO_3_)_3_·9H_2_O, H_2_BDC and ultrapure water. After 30 min of stirring at room temperature, the mixture was transferred to a 25-mL Teflon-lined stainless steel autoclave and incubated at 150 °C for 5 h. An external magnet was used to separate the synthesized Fe_3_O_4_@MIL-53(Al). Then it was washed with DMF, acetonitrile, and ultrapure water, and then dried under vacuum at 60 °C. Finally, the product was calcined in a tube furnace at 300 °C for 24 h.

### 3.5. MSPE Procedure

First, 25 mg of Fe_3_O_4_@MIL-53(Al) was added to 1.5 mL of the sample solution. The mixture was vibrated at 1000 rpm for 5 min and Fe_3_O_4_@MIL-53(Al) was collected using magnetic separation. Then, 1.0 mL of acetonitrile was used to elute the target from the magnetic material (0.5 mL each time). The eluent was filtrated through a 0.22-μm PTFE strainer and 10 μL of the eluent was injected into the HPLC system for analysis (Figure 1).

### 3.6. Methodology Validation

The analytical performance of the method was evaluated using several characteristic parameters, such as limit of detection (LOD), limit of quantification (LOQ), linearity, and relative standard deviation (RSD). The linear range was from 0.05 to 5 μg/mL. The LODs were defined based on signal-to-noise ratios(S/N) of 3, and LOQs were defined at S/N of 10. We evaluated the precision of the method by analyzing the repeatability of the inter-day and intra-day recovery rates, including five repeats in one day and repeats in three days.

## 4. Conclusions

In this study, the magnetic composite Fe_3_O_4_@MIL-53(Al) was successfully synthesized by a simple hydrothermal method. Combined with HPLC technology, the utility of Fe_3_O_4_@MIL-53(Al), as an effective MSPE adsorbent, for four pollutant residues in milk, was verified. The material exhibited good magnetic properties, extraction efficiency, and repeatability, along with good reusability after 10 adsorption and desorption cycles. Moreover, it could adsorb the target within 5 min. In short, the raw materials used in the developed method were inexpensive and could, simultaneously, achieve high efficiency and rapid detection of multi-pollutants in food samples.

## Figures and Tables

**Figure 1 molecules-27-02088-f001:**
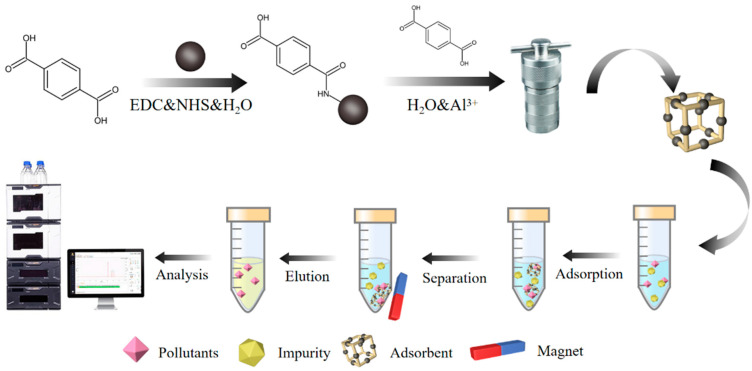
Schematic diagram for the synthesis of Fe_3_O_4_@MIL-53(Al) and MSPE process.

**Figure 2 molecules-27-02088-f002:**
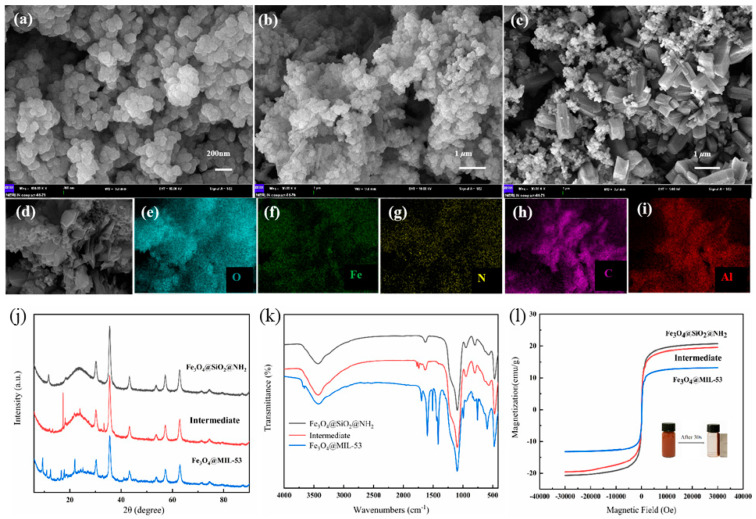
SEM images of amino modified Fe_3_O_4_ magnetic material (**a**), H_2_BDC modified Fe_3_O_4_ magnetic material (**b**), andFe_3_O_4_@MIL-53(Al) (**c**–**i**) SEM-mapping of Fe_3_O_4_@MIL-53(Al). XRD patterns (**j**), FT-IR spectra (**k**), and magnetic hysteresis curves hysteresis (**l**) of magnetic materials.

**Figure 3 molecules-27-02088-f003:**
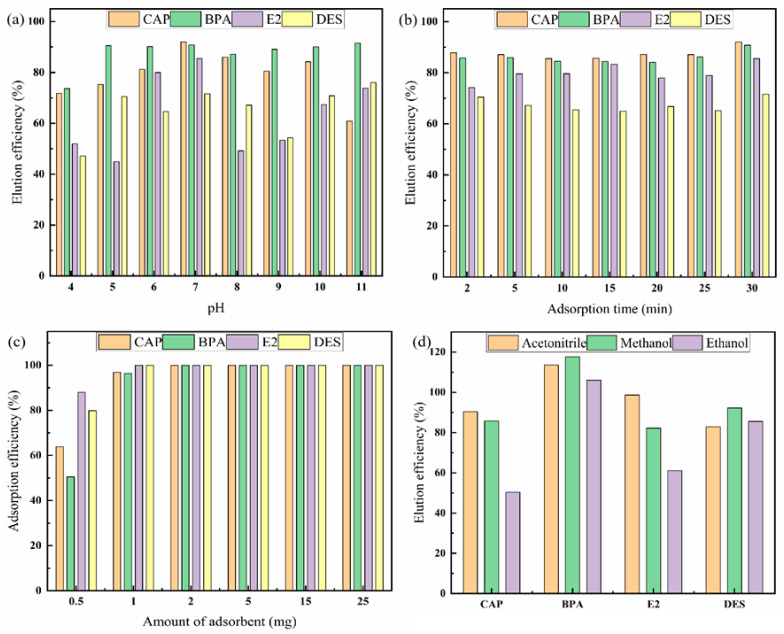
Effect of sample pH (**a**), adsorption time (**b**), the amount of adsorbent (**c**) and elution solvent (**d**) on the recoveries of CAP, BPA, E2, and DES.

**Figure 4 molecules-27-02088-f004:**
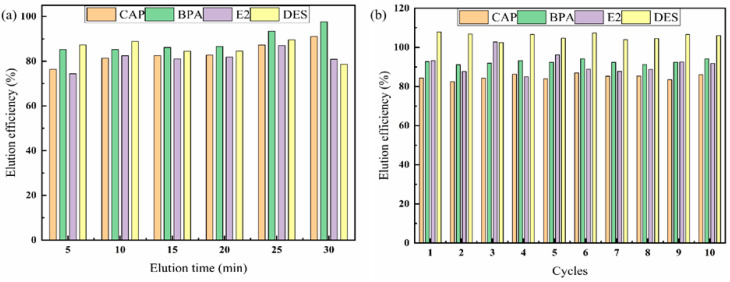
Effect of elution time (**a**) on the recoveries of CAP, BPA, E2, and DES. The reusability of Fe_3_O_4_@MIL-53(Al) (**b**).

**Table 1 molecules-27-02088-t001:** Application and recoveries in milk sample.

	Whole Milk			Skimmed Milk	
Analytes	Added (μg/mL)	Recovery (%)	RSD (n = 3, %)	Recovery (%)	RSD (n = 3, %)
CAP	0	ND		ND	
	0.10	101.49	0.025	92.45	0.054
	0.15	93.43	1.951	97.43	0.053
	0.20	94.78	0.076	88.17	0.011
BPA	0	ND		ND	
	0.10	99.66	0.048	96.52	0.033
	0.15	104.30	0.173	103.88	0.149
	0.20	111.64	0.046	91.94	0.041
E2	0	ND		ND	
	0.10	91.32	0.067	91.93	0.052
	0.15	98.73	0.022	96.90	0.036
	0.20	91.40	0.073	100.41	0.108
DES	0	ND		ND	
	0.10	113.46	0.019	107.58	0.021
	0.15	99.86	1.404	97.63	0.043
	0.20	102.19	0.002	99.40	0.027

**Table 2 molecules-27-02088-t002:** Comparison of proposed method wit reported methods.

Adsorbent	Method of Extraction	Analysis	Target	Linear Range (ng mL^−1^)	LODs (μg L^−1^)	Ref.
Fe_3_O_4_-NC	/	sensor	DES ^a^, E2 ^b^	0.01–20 μmol/L	4.6–4.9 nmol/L	[38]
Fe_3_O_4_/GO/DEHPA NC	MSPE	HPLC-UV	ph ^c^, MP ^d^, PP ^e^, BPA ^f^	0.05–5	2.5–14.3	[39]
MILs	DLLME	HPLC	E1 ^g^, E2, HP ^h^, CMA ^i^, MGA ^j^, MPA ^k^	20–1000	5–15	[40]
MI-MNP	d-SPE	HPLC-UV	BPA	50–1000	0.3	[17]
Fe_3_O_4_@MIL-53(Al)	MSPE	HPLC-UV	CAP ^l^, BPA, E2, DES	50–5000	4–108	This work

^a^ DES: Diethylstilbestrol; ^b^ E2: Estradiol; ^c^ ph: phenol; ^d^ MP: methyl paraben; ^e^ PP: propyl paraben; ^f^ BPA: Bisphenol A; ^g^ E1: estrone; ^h^ HP: 17-α-hydroxyprogesterone; ^i^ CMA: chloromadinone 17-acetate; ^j^ MGA: megestrol 17-acetate; ^k^ MPA: medroxyprogesterone 17-acetate; ^l^ CAP: Chloramphenicol.

## Data Availability

All data generated during the current study are included in this article and as Appendix A.

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
