# Peer review of "Fabrication of Magnetic Al-Based Fe3O4@MIL-53 Metal Organic Framework for Capture of Multi-Pollutants Residue in Milk Followed by HPLC-UV"

_molecules, 2022, doi:10.3390/molecules27072088_

Round 1

Reviewer 1 Report

In this work, magnetic Al-Based Fe3O4@MIL-53 MOFs was fabricated as the adsorbent for capture of multiple pollutants in milk. However, there are some points should be addressed before accepted.

1. Please check the formatting of the manuscript. There are many cases when blank space is not be used and subscripted/superscripted text does not appear correctly.

2.The upper and lower subscripts use non-standard Fe3O4 such as lines 151, 152 and 157.

3. Resolution of various figures is low and can be improved. For example, the size of figure 3 and figure 4 is too small to be readable. Please check all the figures and improve as well as possible.

4. The extraction mechanism should be discussed in more detail.

5. Differences between whole milk and skimmed milk as real samples can be mentioned in the work.

6. Please explain why chloramphenicol (CAP), bisphenol A (BPA), estradiol (E2), and hexadiol (DES) were used as model analytes.

7. There are several grammatical errors in the manuscript.

Reviewer 2 Report

The article is devoted to the actual theme of magnetic MOFs, is done at a high level, well written and can be published after technical editing.

Page 3 line 116
filtered.850 μL - space skipped
Page 4 line 121
and 250 ml ultrapure water. 38 mL NH3 H2O and 50 mL TEOS were added into it.
At which rate were the reagents added? Simultaneously or sequentially? This may be important for particle properties and morphology.
P4L124
functionalization.154 mL space skipped
P4L152, 153
Fe3O4@MIL-53(Al) subscripts should be indicated
P5L172
When 2θ =30.12°ï¼Œ35.6°,43.2°,53.4°,57.1 172 and 62.7°, spaces skipped
P5L183 and below:
When specifying the dimension cm-1, it is required to use a superscript for the degree
P6Fig3, P7Fig4
You should make the pictures larger, in this format their content is hard to read

Reviewer 3 Report

The paper is well written and presents important model results on the effective extraction and analysis of pollutants in natural samples. However, the necessity of such complex adsorbent with a multistep synthetic scheme is questionable. I suppose it was possible to use Fe-MIL-53 or another stable magnetic MOF for MSPE in similar conditions and thus to avoid the magnetic grafting procedure at all. So, the design idea of the reported composite should be disclosed more clearly. 

And in the second, authors need to prove the successful synthesis of MIL-53 phase by adding its theoretcial PXRD pattern to the figure 2j. 

Round 2

Reviewer 1 Report

accept